# Advancing Lithium Battery Performance through Porous Conductive Polyaniline-Modified Graphene Composites Additive

**DOI:** 10.3390/nano14060509

**Published:** 2024-03-12

**Authors:** Hao-Tung Lin, Eunice Chuang, Sheng-Chun Lin

**Affiliations:** 1Energy Storage Laboratory, Industrial Technology Research Institute, Hsinchu 31041, Taiwan; 2CHH LEE Enterprise Co., Ltd., Taichung 434011, Taiwan; unice782001@gmail.com (E.C.); caloslinasus@gmail.com (S.-C.L.)

**Keywords:** polyaniline, graphene, composite, lithium battery

## Abstract

This study aimed to enhance lithium battery performance through the utilization of porous conductive polyaniline-modified graphene composites (PMGCs). Given the growing importance of green energy, coupled with the development of lithium-ion battery systems and electric vehicles, achieving high-speed charge and discharge performance is imperative. Traditional approaches involve incorporating additives like carbon nanotubes and graphene into electrodes to improve conductivity, but they encounter challenges related to cost and aggregation issues. In this study, polyaniline (PANI), a cost-effective, stable, and conductive polymer, was explored. PMGCs was formed by employing ammonium persulfate (APS) as an oxidant during PANI polymerization, simultaneously serving as a surface modifier for graphene. This study systematically investigated the impacts of varying amounts of PMGCs on lithium-ion battery electrodes by assessing the reductions in internal resistance, aging effects, different charge and discharge rates, and cycle performance. The PMGC exhibited a porous structure formed by nanoscale PANI intertwining on graphene. Various measurements, including FT-IR, TGA, Raman spectroscopy, and battery performance assessments, confirmed the successful synthesis and positive effects of PMGCs. The results indicated that a 0.5% addition of PMGC led to a reduced internal resistance and enhanced fast-charge and discharge capacity. However, an excessive amount of PMGCs adversely affected aging and self-discharge. This study provides valuable insights into optimizing the PMGC content for improved lithium battery performance, presenting potential advancements in energy storage systems and electric vehicles.

## 1. Introduction

In efforts to reduce environmental pollution, there is a significant emphasis on green energy, leading to the flourishing development of lithium-ion battery energy storage systems and electric vehicles. To meet the fast charging demands for electric vehicles and the automatic frequency regulation capabilities required for energy storage systems to provide rapid response reserve services, high-speed charging, and discharging performances have become essential requirements for future lithium-ion batteries. To enhance the fast charging and discharging performances of lithium-ion batteries, it is most common to employ methods that reduce the internal resistance of the battery. Owing to their high conductivity, carbon nanotubes and graphene are frequently employed to modify the active materials of anodes and cathodes and to serve as conductive additives in electrodes. Modifying active materials and using conductive additives expedite electron conduction between active materials and between active materials and current collectors. Furthermore, these methods reduce the overall electrode contact resistance, thereby notably alleviating polarization phenomena, especially during high-current-density charge and discharge processes.

Wang et al. [1] introduced a coating layer of single-walled carbon nanotubes (SWNTs) to enhance the battery performance of conversion-type anode materials. This SWNT layer significantly boosts the electrical conductivity of active materials and facilitates the free transport of Li^+^ ions from the electrolyte to Fe_3_O_4_ due to its mesoporous structure. Liu et al. [2] studied the use of solid-state mechanochemical methods for fabricating typical graphene materials for LIB anodes and indicated that mechanochemical tools hold potential and effectiveness in exfoliating 2D materials and, forming composites with metal oxides in addition to doping them with heteroatoms to enhance their lithium storage capability. Choi et al. [3] incorporated carbon nanotubes (CNTs) as additives in conjunction with LiNi_0.8_Co_0.1_Mn_0.1_O_2_ cathode material. The outstanding cycling stability achieved with CNTs at 5C conditions is ascribed to their significantly reduced polarization compared to that of conventional carbon black (Super P) cells. Shi et al. [4] explored the impact of graphene, prepared through three typical methods, on the electrochemical characteristics of LiCoO_2_-based cathodes. It was observed that graphene possessing oxygen functional groups, a relatively low surface area, and a suitable size demonstrated improved electrochemical performance when utilized as a conductive additive. Furthermore, when integrated with Super P, a mere 1% of graphene proved sufficient for establishing an effective conductive network within the electrode.

When utilized as conductive additives, carbon nanotubes (CNTs) exhibit a small diameter at the nanometer scale and a high aspect ratio (>1000), resulting in an exceptionally large surface area. Similarly, graphene also possesses a notably large surface area. In addition to high production costs, the substantial surface area of these entities leads to an increased tendency toward agglomeration. The dispersion of these two materials also presents a significant challenge. Choi et al. [3] employed ultra-sonication during the slurry preparation process to achieve the optimal dispersion of carbon nanotubes. Perumal et al. [5] introduce numerous methods that have been used to address the issue of agglomeration, including mechanical exfoliation and the use of solvents, surfactants, and polymers. Liu et al. [2] introduced graphene composites such as ZnO/graphene, Fe_3_O_4_/graphene, and SnO_2_/graphene which were obtained through mechanochemistry. Bordes et al. [6] introduced the dispersion and stabilization of exfoliated graphene in an ionic liquids, which is a solvent-based method used to disperse graphene. Luo et al. [7] proposed a strategy utilizing graphene oxide (GO) to effectively disperse graphene, aiming to produce homogeneous, adjustable, and high-concentration graphene dispersions. It was noted that the structure of GO, with abundant edge-bound hydrophilic carboxyl groups and in-plane hydrophobic π-conjugated domains, allows it to function as a special ‘surfactant’ that enable graphene dispersion. Ma et al. [8] highlighted that the proper dispersion of CNTs and appropriate interfacial adhesion between CNTs and the polymer matrix were essential. Perumal et al. [5] reported that employing polymers for graphene dispersion presents advantages compared to utilizing surfactants and solvents. Polymer functionalization offers the benefits of altering the molecular weight and topological structure, thereby enabling the selection of polymers that are suitable for the application. However, achieving stable graphene dispersions using polymers is challenging due to the difficulty of selecting a suitable polymer.

Polyaniline, derived from the monomer aniline, is a conductive polymer that is commonly employed as an electrode material in supercapacitors [9]. Polyaniline is characterized by its low cost, low density, high stability in oxidation and water, electrical conductivity, and reversible oxidation and reduction. It is also applied as an active material for both the cathode and anode in lithium batteries [9,10,11]. Depending on the degree of oxidation, a single polyaniline polymer chain may exhibit both a reduced benzene–benzene state and an oxidized benzene–quinone state [12]. In each case, adjacent benzene monomers are connected by amino and imine groups. The energy gap of polyaniline (~1.54 eV) is similar to the semiconductor energy gap (~1.3 eV). Through a doping process, the energy gap can be reduced to enhance conductivity. In an acidic environment, the nitrogen in the imine group in polyaniline is easily protonated. It exhibits conductivity due to the conjugated double-bond structure in the main chain of polyaniline.

F. Zeng et al. [13] found that the microstructures of polyanilines formed via interfacial polymerization with a single oxidant (ammonium persulfate) were markedly different from those formed with composite oxidants like APS/FeCl_3_ (ammonium and ferric chloride) or APS/K_2_Cr_2_O_7_ (ammonium and potassium dichromate). When introducing FeCl_3_ as the second component instead of using APS as a single oxidant, a similar nanofibrous morphology in PANI was observed, with a slight increase in the average diameter from 70 nm to 80 nm. However, PANI prepared with K_2_Cr_2_O_7_ as the second component in composite oxidants did not exhibit the typical nanofibrous structure seen in general interfacial polymerization systems. Specifically, the product displayed a unique petal-like structure with sphere diameters of 500 nm. Beyond the essential parameters for aniline oxidation, in accordance with Tran et al. [14], it was found that various other reaction and processing parameters influence the microstructure of polyaniline, including the type of template employed, the introduction of chemical additives during aniline oxidation, and the specific methodology utilized in the polymerization process. According to Felipe de Salas et al. [15], the microstructure of polyaniline was significantly influenced by external templates guiding nanostructural growth either within or around self-assembled micelles. The use of dodecylbenzenesulfonate (SDBS) led to nanofibers, docusate sodium salt (AOT) resulted in granular spheres, sodium dodecyl sulfate (SDS) produced splinters, and sodium lauryl ether sulfate (SLES) led to amorphous agglomerates. Sathish et al. [16] achieved the oxidative polymerization of aniline by applying MnO_2_ to the surface of graphene, leading to the formation of a microstructure comprising porous polyaniline (PANI) nanofibers on the graphene surface.

Previous studies have rarely explored the uniform dispersion of nanoporous polyaniline on the graphene surface to form graphene–polyaniline composites. In this study, hydrochloric acid (HCl) was used as a dopant, and ammonium persulfate was employed as the oxidant during the polyaniline polymerization process. Additionally, it served as a surface modifier for graphene, resulting in the formation of polyaniline-modified graphene composites. The uniform dispersion of nanoporous polyaniline aids in the adsorption of electrolytes, thereby facilitating ion diffusion within the electrode and reducing internal resistance. The literature has indicated that polyaniline–graphene composites were frequently employed in supercapacitors [9,17,18], with comparatively fewer applications as conductive additives for electrodes in lithium-ion batteries. These composites are characterized by low production costs and excellent dispersibility during slurry preparation, addressing the aforementioned issues simultaneously. In this report, PMPC was used as a conductive additive for the electrodes of lithium-ion batteries. We measured DCIR, impedance, and the effects of varying amounts of PMGC on reducing the internal resistance. Furthermore, we assessed its impact on the charging and discharging efficiencies and the cycling performance of the battery.

## 2. Material and Methods

### 2.1. Preparation of Polyaniline-Modified Graphene Composites

Polyaniline-modified graphene composites were synthesized using aniline (99.5% Thermo Scientific, Waltham, MA, USA) and graphene (Enerage, Yilan, P-ML20, Taiwan) as primary materials. A hydrochloric acid solution (37%, Honeywell, Fluka, Charlotte, NC, USA) was diluted with deionized water and utilized as the doping agent. Ammonium persulfate (98%, Showa, Yamanashi, Japan) was dissolved in deionized water and employed as the oxidant. The synthesis process involved the following steps: initially, graphene was added to aniline under a nitrogen atmosphere, and the mixture underwent oscillation using a bath-type ultrasonic oscillator at a low temperature (≤15 °C) for 6 h. Deionized water was added to the glass-jacketed reaction vessel, and then, cooling was performed. Subsequently, the pre-mixed graphene–aniline slurry was introduced into the reaction vessel, and diluted hydrochloric acid was added. The aqueous ammonium persulfate solution was slowly dripped into the vessel at approximately 5 °C. The resulting precipitate was collected and, dried under a vacuum, and the doped material was obtained. This material underwent de-doping through the introduction of ammonium hydroxide, resulting in the formation of polyaniline-modified graphene composites.

### 2.2. Characterization

The PMGC powder was compressed into 20 mm discs at a pressure of 30 MPa, and then its intrinsic electrical conductivity was measured using a four-point probe. The resulting measured value was approximately 0.08 Siemens according to the CT 5601Y CT 5601Y sheet resistivity meter (Quatek Co., Ltd., Taipei, Taiwan).

The material’s microstructure was examined via field emission scanning electron microscopy (ZEISS ULTRA PLUS, Hitachi S-3000N, Tokyo, Japan). The tap density was determined using Micromeritics ASAP 2020. A thermogravimetric analysis (TGA) was conducted with NEXTA STA200 by Hitachi to observe the thermal weight loss behavior of the composite. Raman spectroscopy (UniDRON, Seul, Republic of Korea) and Fourier-transform infrared spectroscopy (FT-IR, PerkinElmer Spectrum 100 FT-IR Spectrometers, Waltham, MA, USA) were employed as primary techniques for studying molecular structures, confirming the presence of polyaniline coating on the surface of graphene.

### 2.3. Battery Fabrication

Regarding the batteries fabricated in this study, the primary materials utilized are detailed as follows. The cathode electrode material comprised NMC622 (T61R, Shanshan Energy Technology Co., Ltd., Shanghai, China), while the anode electrode material was a blend of intercalated graphite microspheres (MCMB) MG11-A (China Steel Chemical Corporation, Kaohsiung, Taiwan), and graphite MAGE3 (Showa Denko Materials Co., Ltd., Tokyo, Japan). The percentage compositions of the anode and cathode electrodes are listed in Table 1. The key components of the electrolyte included ethylene carbonate (EC), ethyl methyl carbonate (EMC), and diethyl carbonate (DEC). The separator (WE20, WSCPOE, Tokyo, Japan) was constructed from polyethylene (PE) and featured a three-layer structure with ceramic layers on both sides, resulting in a total separator thickness of 16 μm.

A pouch-type lithium battery with a 3600 mAh capacity was produced, as depicted in Figure 1, featuring electrode dimensions approximately measuring 120 mm in length and 90 mm in width. The standard battery manufacturing process comprised battery design, electrode fabrication, battery assembly, and formation. Following the completion of battery assembly, and before commencing the formation process, the internal resistance of the battery was measured using a battery internal resistance Tester (Hioki BT3563, Ueda, Japan).

As indicated in Table 2, an evaluation of the influences of varying PMGC quantities was conducted by incorporating 0%, 0.5%, and 1% of PMGCs into the cathode and anode electrodes, respectively. These additions replaced a minor portion of the active material within the electrodes. The cathode electrode and anode electrode were designated three distinct codes each: Ca1, Ca2, and Ca3 and An1, An2, and An3, respectively.

The cathode and anode electrodes, with varying contents (as outlined in Table 2), were paired and assembled into complete batteries. The nomenclature of the batteries in Test 1 is detailed in Table 3. Test 1 encompassed 9 groups (G1-G9) of batteries, each consisting of 5 batteries. G1 functioned as the control group without PMGCs, employing Ca1 and An1 for the cathode and anode electrodes, respectively. The five batteries within the control group were labeled from C0A0-1 to C0A0-5. The nomenclature for battery IDs in the other groups (G2-G9) followed a similar sequence.

Drawing insights from the experimental findings of Test 1, the optimal quantity of the PMGC was identified. Subsequently, Test 2 was formulated, as outlined in Table 4. The primary distinction between Test 1 and Test 2 resided in the density of the anode electrodes, with the former being 1.4 g/cc and the latter being 1.57 g/cc. The anode electrode code for the latter was designated NAn1, and it did not incorporate PMGCs. This decision was informed by the outcomes of Test 1, where the addition of PMGCs into the anode electrode did not yield favorable results.

The cathode and anode electrodes, featuring diverse compositions detailed in Table 4, were paired and assembled into complete batteries. The nomenclature of the batteries in Test 2 is provided in Table 5. Test 2 comprised 3 groups (G10, G11, and G12) of batteries, each consisting of 4 batteries. G10 functioned as the control group without a PMGC.

## 3. Results and Discussion

### 3.1. Characteristics of Polyaniline-Modified Graphene Composites

The powder of the synthesized polyaniline-modified graphene composites, exhibiting a measured tap density of 1.43 g/cm³, offers advantages for slurry preparation and energy density in comparison to typical carbon black, graphene, and CNT, all of which possess tap densities of less than 1 g/cm³. Figure 2a,b depict pure graphene and PMGC, respectively, with porous polyaniline visibly covering the graphene. In Figure 2c, polyaniline, approximately 10 nanometers in width, forms a network of nano-sized pores, coating the surface of graphene. The addition of these PMGCs to the battery electrode led to speculation that these nanoscale pores would enhance the absorption and infiltration of the lithium battery electrolyte.

Figure 3 presents the FT-IR spectra of graphene and polyaniline-modified graphene composites. In comparison to PMGC, the spectrum of graphene does not reveal distinct features. However, for PMGC, peaks at 1576 and 1490 cm^−1^ are attributed to the stretching vibration of the quinonoid and benzenoid rings, respectively [19]. Additionally, bands at 1294 cm^−1^ can be assigned to the π-electron delocalization induced in the polymer through protonation or C-N-C stretching vibration [19]. Peaks at 1239 cm^−1^ and 1133 cm^−1^ are ascribed to C–N and C–H bending of benzenoid and quinoid rings [20,21]. Figure 3 showcases the characteristic absorption bands of polyaniline, confirming the successful synthesis of polyaniline, which was coated on the surface of graphene.

The weight loss of the composite with a temperature variation can be seen in Figure 4. Primary weight loss is characterized by three distinct steps. The initial stage of weight loss, approximately 6%, occurs from room temperature to around 100 °C and is attributed to the vaporization of moisture and the elimination of unreacted monomers in liquid form. The following stage, occurring between 100 °C and 300 °C, is presumed to be linked to the degradation of doped acid and oligomeric aniline compounds generated throughout the reaction, resulting in an approximate weight loss of 10% (94–84%). Zeng et al. [22] reported that the dopant, HCl, was almost completely removed from the polyaniline below 250 °C. The final stage, occurring at temperatures greater than 300 °C, could be ascribed to the complete decomposition of the polymer backbone [23,24]. 

Figure 5 depicts the Raman spectra of graphene and polyaniline-modified graphene composites. Graphene exhibits a prominent Raman-active peak at 1576 cm^−1^, corresponding to the G band related to the first-order scattering of the E2g mode observed for sp2-carbon domains [25]. The Raman-active peak at 1350 cm^−1^, assigned to the D band corresponding to structural defects, is not prominently observed. In the PMGC curve, C–H bending deformation in the benzenoid ring at 1168 cm^−1^, C–N+ stretching at 1335 cm^−1^, C=N stretching vibration at 1485 cm^−1^, and C=C stretching of quinoid at 1585 cm^−1^ are evident [21,26], indicating the presence of the polyaniline structure on the surface of graphene. This outcome aligns with the results obtained from the FTIR analysis.

### 3.2. Internal Resistance

All assembled batteries underwent internal resistance testing prior to the formation process. Batteries with the addition of 0.5% of PMGCs in the cathode electrode, labeled from C0.5A0-1 to C0.5A0-5 (G2), exhibited reductions in internal resistance compared to the control group, C0A0 (G1), as depicted in Figure 6. The average resistance decreased from approximately 11.2 mΩ for G1 to an average of 8.3 mΩ for G2. However, the introduction of 1% of PMGCs in C1A0 (G3) resulted in an average resistance of 9.2 mΩ, showing no further decrease compared to 0.5% of PMGCs. This observation was speculated to be associated with the directional nature of graphene. Liu et al. [27] reported that an LFP with small-size graphene (GN) shows better specific capacity and rate performance than those with medium-size and large-size GN, and the LFP with large-size GN displays a poor rate performance. Given graphene’s 2D structure, a longer ion transmission path in the plane may lead to inefficient ion conductivity when a higher quantity of PMGC is added.

Nevertheless, the inclusion of PMGCs in the anode electrode led to an increase in the internal resistance, as evident in C0A0.5 (G4) and C0A1 (G7). This was attributed to the lower electrical conductivity of polyaniline in the PMGC, which replaced some of the higher electrical conductivity graphite. According to Atiqah et al. [28], the electrical conductivity of graphite (0.200 S/cm) is approximately 12 times that of pure PANI (0.017 S/cm). The addition of PMGCs to the anode electrodes resulted in polyaniline having comparatively less advantageous conductivity in the composite.

### 3.3. Aging

To evaluate the self-discharge behavior, an aging test was conducted on the batteries. Following charging to 4.2 V, the batteries were left at room temperature for 24 h, and their voltages were tested. Figure 7 illustrates the voltage drop after 24 h for C0.5A0 (G2), which contained 0.5% of PMGCs. Compared to the control group, C0A0 (G1), a significant difference was not observed. However, batteries with 1% of PMGCs in either the cathode or anode electrode, such as C1A0 (G3), C1A0.5 (G6), C0A1 (G7), C0.5A1 (G8), and C1A1 (G9), exhibited noticeable voltage drops after 24 h of aging. This suggests that a higher proportion of PMGCs may contribute to significant self-discharge. The excessive addition of conductive additives could have led to negative effects, as they were less likely to disperse uniformly, potentially resulting in self-discharge.

### 3.4. Fast-Charge Performance

At room temperature, constant current (CC) charging was performed with different currents (1C, 2C, 3C, and 5C) until they reached 4.2 V, followed by constant voltage (CV) charging, which was carried out until the current reached 0.2C. The charge capacity during constant current charging (CC) was observed. In Figure 8, the solid line represents the C0.5A0 battery (G2) with 0.5% of PMGCs in the cathode electrode, and the dashed line represents the control group batteries (C0A0 (G1)) without PMGCs in the cathode electrode.

The charging capacities at constant currents of 2C, 3C, and 5C were divided by the charging capacity at a constant current of 1C, which revealed noteworthy findings. The battery incorporating 0.5% of PMGCs exhibited a substantial enhancement in its 3C CC charging capacity, achieving an impressive 80% of the baseline. This outperformed the control group battery, which lacked PMGCs and registered a lower charging capacity of 60% of the baseline. The utilization of PMGCs evidently contributes to a more efficient and rapid charging process, showcasing this method’s potential to significantly improve the overall performance of a battery system, as illustrated in Table 6.

At room temperature, discharge capacity testing was performed with charging conditions involving 1C constant current (CC) charging until it reached 4.2 V, followed by constant voltage (CV) charging until the current reached 0.2C. Subsequently, the discharge capacity was measured under discharge conditions at different currents (1C, 2C, 3C, 4C, and 5C) until the current reached 2.75 V.

The experimental results presented in Figure 9 indicate that the discharge capacity of C0.5A0 (G2) is slightly better than that of C0A0 (G1). Using the capacity at 1C as a reference point, the retention percentages for capacities of 2C, 3C, 4C, and 5C were calculated by dividing each capacity by the 1C capacity. The actual values, shown in Table 7, indicate that the difference in performance between the two is not very significant at high C rates.

### 3.5. DCIR and EIS

To evaluate the stability of the battery, the battery at a 30% state of charge (SOC) was stored at room temperature for 6 months, after which its DCIR (direct current internal resistance) and EIS (electrochemical impedance spectroscopy) were measured.

The DCIR testing procedure was as follows: In Step 1, the battery was charged to 4.2 V. In Step 2, a discharge process was conducted at 0.03 A for 30 min, and the voltage value at the last second of the discharge was recorded as V1. Subsequently, a discharge process was performed at 6 A for 5 s, and the voltage value at the last second of the discharge process was recorded as V2. The DCIR at 100% SOC was then calculated using Formula (1), where I1 was 0.03 A and I2 was 6 A. Following that, a discharge process was carried out at 0.6 A for 30 min, followed by a 30-s rest period. In Step 3, the process of Step 2 was repeated nine times, and the DCIR was calculated at 90%, 80%, and down to 10% SOC. The DCIR result, depicted in Figure 10, highlighted a lower DCIR in the battery denoted as C0.5A0, where 0.5% of PMGCs was added to the cathode electrode compared to the control battery, C0A0, without the additive.
DCIR = (V_1_ −V_2_)/(I_2_ − I_1_)(1)

The EIS was conducted using the BioLogic SP-300 equipment with a frequency range spanning from 10 mHz to 0.5 MHz. Figure 11 shows the representative impedance spectra measured around 3.6 V from the C0A0 and C0.5A0 batteries. Electrochemical impedance spectra can be represented as Nyquist plots. The Nyquist plot is a graphical representation of the real part of impedance (Zreal) and the imaginary part of impedance (Zimage) plotted at different angular frequencies. Re (equivalent series resistance) accounts for the pure ohmic resistance of electrodes, electrolytes, connecting wires, and current collector foils. Rct represents the charge transfer resistance, and its value can be estimated from the semicircle. The value of Re was determined by the intersection point of the Nyquist plot with the X-axis (Re(Z)), with the detection frequency in the high-frequency region [29,30]. The test results indicated that, compared to C0A0 with Re (0.043 Ω) and Rct (0.028 Ω), C0.5A0 showed lower values for both Re (0.028 Ω) and Rct (0.018 Ω).

### 3.6. Fast Discharge and Cycle Performance

To simulate higher energy density for the rechargeable battery, experiments were designed for Test 2. The anode electrode density of the batteries in Test 1 was 1.4 g/cc, while it was 1.57g/cc for the batteries in Test 2. The former has a higher porosity in the electrode, allowing the electrolyte to wet the pores and enhance ion conductivity. Consequently, batteries with the same additive content in Test 1 exhibited lower impedance compared to those in Test 2. Compared with the internal resistance in Test 1, the average resistance for C0A0 (G1) was 11.2 mΩ, while in Test 2, the average resistance for NC0A0 (G10) was 29.7 mΩ. For C0.5A0 (G2), the average resistance was 8.3 mΩ, whereas that of NC0.5A0 (G11) was 12.3 mΩ. Similarly, C1A0 (G3) had an average resistance of 8.8 mΩ, while that of NC1A0 (G12) was 16.3 mΩ. This difference could be attributed to the higher porosity and ion conductivity in the anode electrodes of the Test 1 batteries.

Compared to the control group (NC0A0 (G10)), batteries incorporating 0.5% of PMGCs (NC0.5A0, G11) and 1.0% of PMGCs (NC1A0, G12) demonstrated markedly reduced internal resistance. Notably, G11 shows an even lower resistance, which is consistent with the results observed in Test 1.

When comparing NC0.5A0 (G11) to NC0A0 (G10), the former exhibits superior discharge capacities at 2C, 3C, 4C, and 5C. Additionally, it demonstrates better discharge percentages at higher C-rates, as depicted in Figure 12 and detailed in Table 8.

Cycling tests were conducted at room temperature with a 1C charging rate and a 1C discharging rate, operating within a voltage range from 2.75 V to 4.2 V. In contrast to NC0A0 (G10) and NC1A0 (G12), the battery featuring 0.5% of PMGCs, NC0.5A0 (G11), showcased the most favorable cycling performance. Figure 13 illustrates that, after 200 cycles, NC0.5A0 (G11) retained 90% of its charge, indicating excellent stability and cycle life for the battery with PMGCs inside the cathode electrode.

## 4. Conclusions

This study investigates the enhancement of lithium-ion battery performance through the utilization of polyaniline-modified graphene composites, with a specific focus on porous morphology. Incorporating polyaniline, a conductive polymer, as a surface modifier for graphene demonstrated effectiveness in reducing the internal resistance and improving the overall battery performance. The distinctive porous nanostructure of polyaniline on graphene was explored to address the challenges associated with additives like carbon nanotubes and graphene, including cost and aggregation issues.

A comprehensive analysis revealed that adding 0.5% of PMGCs to the cathode electrode significantly reduced internal resistance, contributing to enhanced battery performance. However, adding 1% of PMGCs did not provide additional benefits, emphasizing the significance of an optimal concentration. This study also investigated how PMGCs influence various electrode densities, unveiling their complex impacts on internal resistance.

The morphology of PMGCs, characterized by porous polyaniline intertwining with graphene, was observed, suggesting their potential benefits in lithium battery electrolyte absorption and infiltration. Various characterization techniques, including FT-IR, TGA, and Raman spectroscopy, confirmed the successful synthesis of PMGCs and their structural features.

The evaluation of fast charging and discharging performances, aging tests, and cycle performance aimed to assess the practical implications of PMGCs. This study demonstrated that adding 0.5% of PMGCs to the cathode electrode enhanced its rapid charging capabilities, resulting in a higher charge capacity compared to that of the control group. Additionally, PMGC-incorporated batteries exhibited lower internal resistance, improved stability, and a favorable cycle life, which were particularly evident in the 0.5% PMGC concentration.

These findings highlight the potential of using polyaniline-modified graphene composites with porous nanostructures as a promising solution to advance lithium-ion battery technologies. This study’s insights contribute to ongoing efforts to develop sustainable and high-performance energy storage systems for applications such as electric vehicles and energy storage systems.

## Figures and Tables

**Figure 1 nanomaterials-14-00509-f001:**
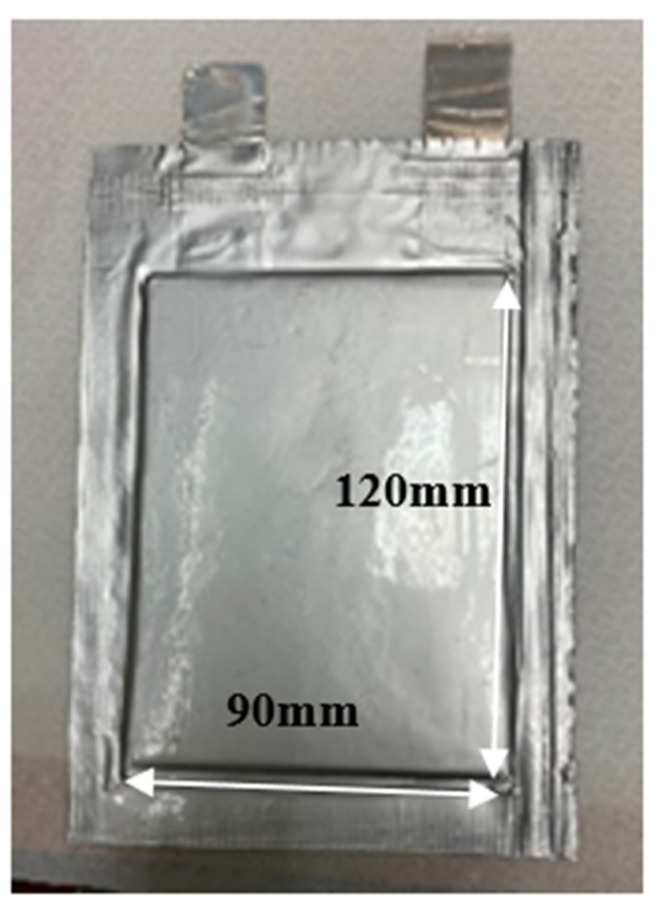
Lithium-ion battery with 3600 mAh capacity.

**Figure 2 nanomaterials-14-00509-f002:**
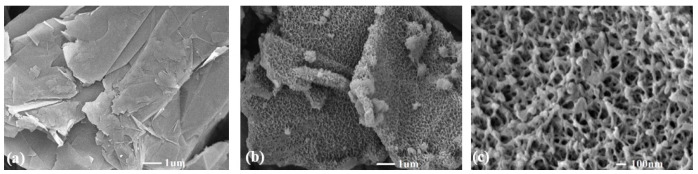
SEM images of (**a**) pure graphene; (**b**) PMGC; (**c**) higher magnification image of (**b**) PMGC.

**Figure 3 nanomaterials-14-00509-f003:**
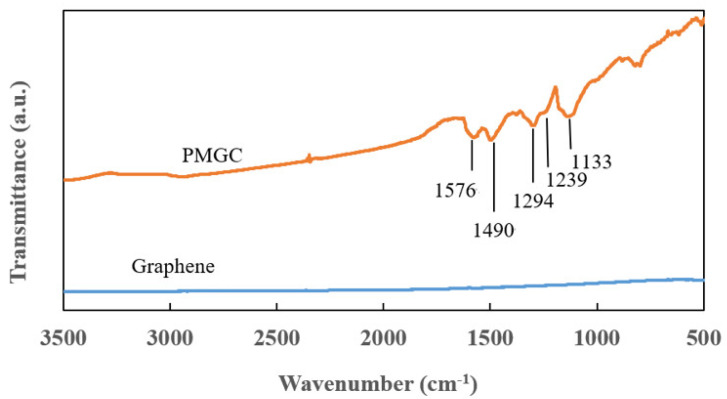
FT-IR spectra of graphene and PMGC.

**Figure 4 nanomaterials-14-00509-f004:**
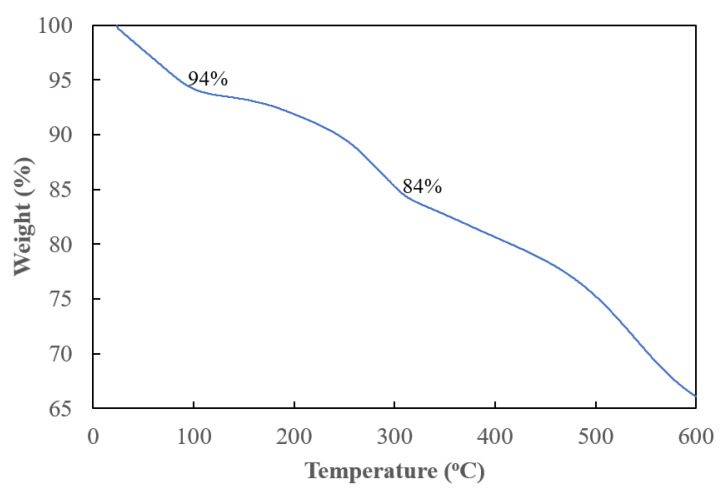
Thermogravimetric curves of polyaniline-modified graphene composites.

**Figure 5 nanomaterials-14-00509-f005:**
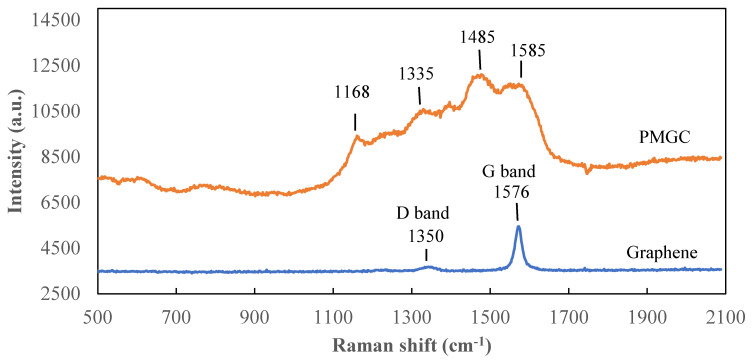
Raman spectra of graphene and polyaniline-modified graphene composites.

**Figure 6 nanomaterials-14-00509-f006:**
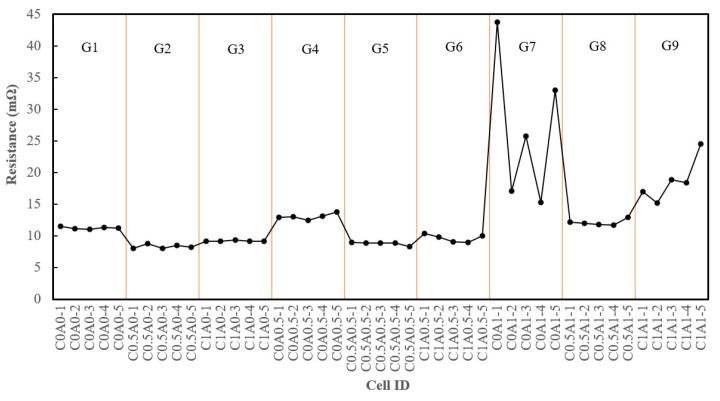
The internal resistance of G1–G9, which was tested prior to the formation process.

**Figure 7 nanomaterials-14-00509-f007:**
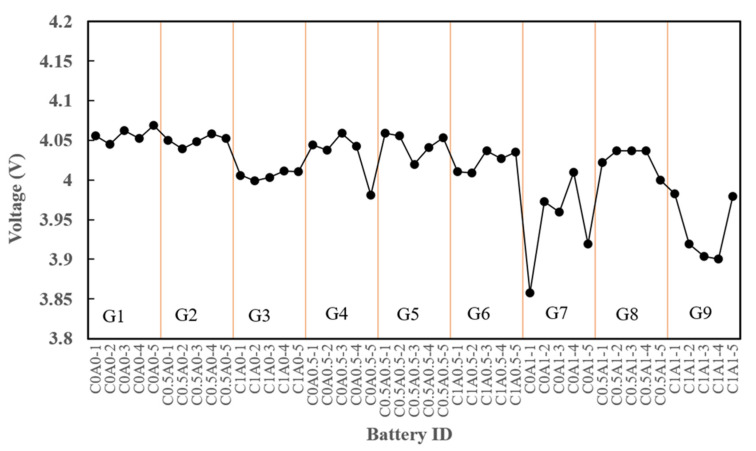
The voltage of G1-G9 after 24 h of aging.

**Figure 8 nanomaterials-14-00509-f008:**
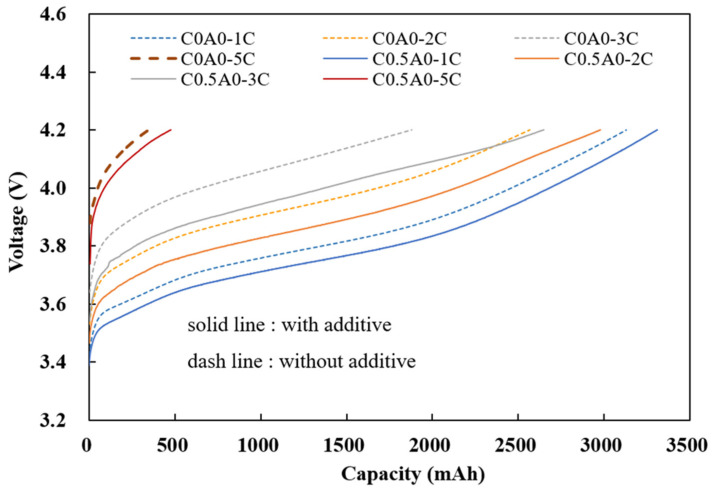
The capacity under different constant current charging conditions.

**Figure 9 nanomaterials-14-00509-f009:**
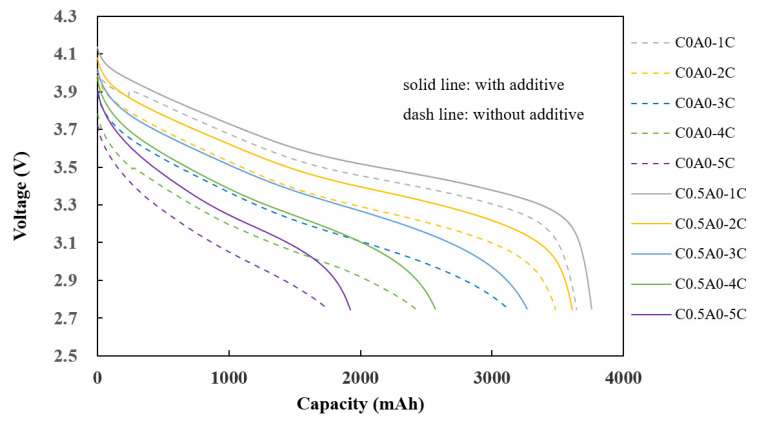
Discharge capacity of batteries in Test 1.

**Figure 10 nanomaterials-14-00509-f010:**
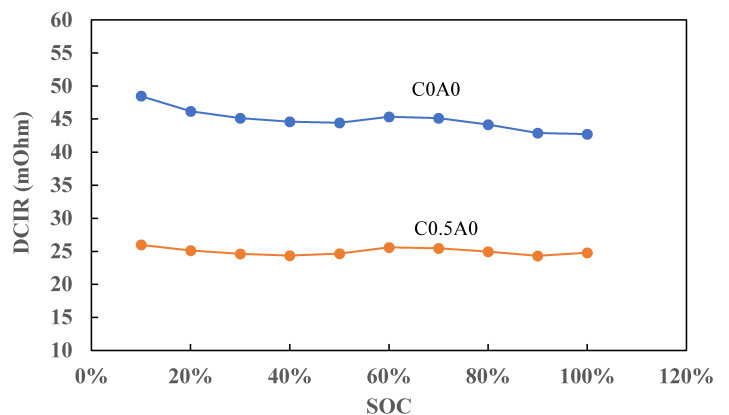
DCIR of C0A0 and C0.5A0.

**Figure 11 nanomaterials-14-00509-f011:**
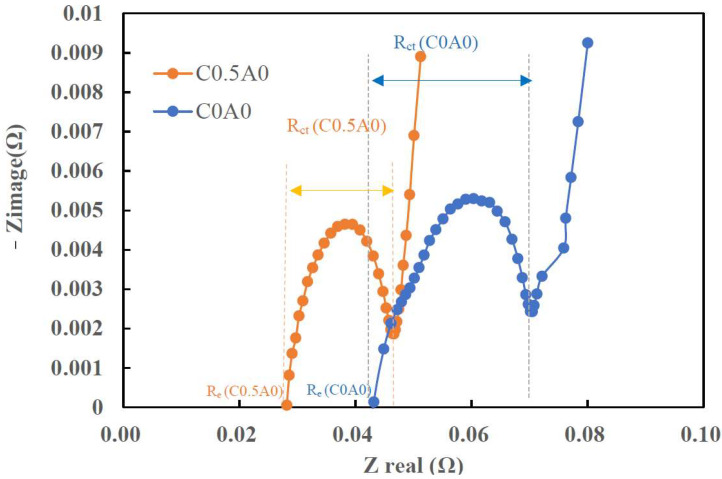
Impedance of C0A0 and C0.5A0.

**Figure 12 nanomaterials-14-00509-f012:**
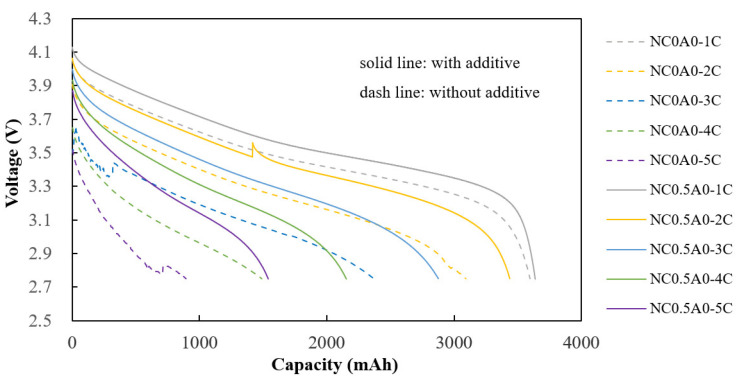
Discharge capacity of batteries in Test 2.

**Figure 13 nanomaterials-14-00509-f013:**
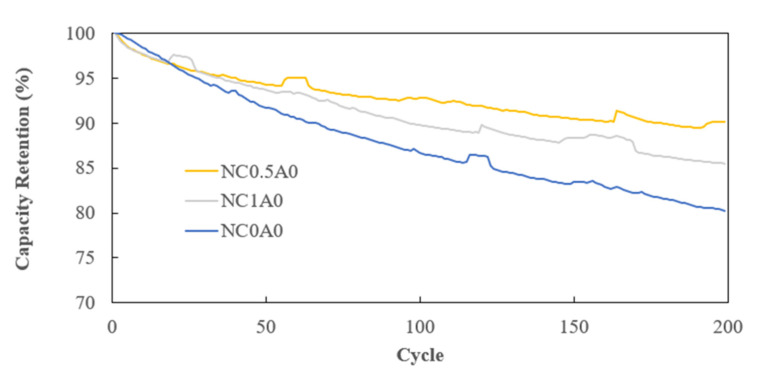
Cycle performance.

**Table 1 nanomaterials-14-00509-t001:** The percentage composition of the electrodes.

Anode	Cathode
95% (active materials + PMGC)	96% (active materials + PMGCs)
2% carbon black (super P)	2% carbon black (super P)
3% binder	2% binder

**Table 2 nanomaterials-14-00509-t002:** Percentage of PMGC added to the electrodes.

Electrode	Code	Active Materials (%)	PMGC (%)
Cathode electrode	Ca1	100	0
	Ca2	99.5	0.5
	Ca3	99	1
Anode electrode	An1	100	0
	An2	99.5	0.5
	An3	99	1

**Table 3 nanomaterials-14-00509-t003:** The electrode codes used in different groups in Test 1.

Group	Battery ID	Cathode Electrode	Anode Electrode
G1	C0A0-1~C0A0-5	Ca1	An1
G2	C0.5A0-1~C0.5A0-5	Ca2	An1
G3	C1A0-1~C1A0-5	Ca3	An1
G4	C0A0.5-1~C0A0.5-5	Ca1	An2
G5	C0.5A0.5-1~C0.5A0.5-5	Ca2	An2
G6	C1A0.5-1~C1A0.5-5	Ca3	An2
G7	C0A1-1~C0A1-5	Ca1	An3
G8	C0.5A1-1~C0.5A1-5	Ca2	An3
G9	C1A1-1~C1A1-5	Ca3	An3

**Table 4 nanomaterials-14-00509-t004:** The percentage of PMGCs in the electrodes of the batteries in Test 2.

Electrode	Code	Active Materials (%)	PMGC (%)
Cathode electrode	Ca1	100	0
	Ca2	99.5	0.5
	Ca3	99	1
Anode electrode	NAn1	100	0

**Table 5 nanomaterials-14-00509-t005:** The electrode codes that were utilized in various groups within Test 2.

Group	Battery ID	Cathode Electrode	Anode Electrode
G10	NC0A0-1~NC0A0-4	Ca1	NAn1
G11	NC0.5A0-1~NC0.5A0-4	Ca2	NAn1
G12	NC1A0-1~NC1A0-4	Ca3	NAn1

**Table 6 nanomaterials-14-00509-t006:** The charging capacity of C0A0 and C0.5A0 under different constant currents.

	1C	2C	3C	5C
C0A0 (mAh)	3130	2567	1879	350
	100%	82%	60%	11%
C0.5A0 (mAh)	3311	2979	2650	476
	100%	90%	80%	14%

**Table 7 nanomaterials-14-00509-t007:** Discharge capacity of C0A0 and C0.5A0.

Battery	1C	2C	3C	4C	5C
C0A0 (mAh)	3643	3483	3124	2421	1749
	100%	96%	86%	66%	48%
C0.5A0 (mAh)	3757	3610	3267	2569	1924
	100%	96%	87%	68%	51%

**Table 8 nanomaterials-14-00509-t008:** Discharge capacity of NC0A0 and NC0.5A0.

Battery	1C	2C	3C	4C	5C
NC0A0 (mAh)	3596	3092	2371	1490	895
	100%	86%	66%	41%	25%
NC0.5A0 (mAh)	3638	3438	2877	2154	1541
	100%	95%	79%	59%	42%

## Data Availability

The data that support the findings of this study are available on request from the corresponding author. The data are not publicly available due to privacy or ethical restrictions.

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
