# Peer review of "Advancing Lithium Battery Performance through Porous Conductive Polyaniline-Modified Graphene Composites Additive"

_nanomaterials, 2024, doi:10.3390/nano14060509_

Round 1
Reviewer 1 Report
Comments and Suggestions for Authors
The manuscript nanomaterials-2855038 ’Advancing Lithium Battery Performance through Porous Conductive Polyaniline-Modified Graphene Composites Additive’ by Hao Tung Lin, Eunice Chuang and Sheng Chun Lin reports how the lithium-ion batteries properties can be improved by the incorporation of porous conductive polyaniline-modified graphene composite additive to the electrodes. PMGC composite was characterised by means of SEM, Raman, FT-IR and TGA analysis. Further the influence of various quantities of PMGC incorporated in the cathode and anode electrodes was evaluated in order to identify the optimal quantity which enhances the battery properties.
The work contained in this article is very interesting, well-structured; analytical approach is robust, but there are some important points that should be addressed before its publication.
Bellow I will discuss my main comments and suggestions which hopefully can help the authors to improve the quality of their work:
1. The Introduction section can be improved by providing a more critical discussion of related literature; presenting recent studies and indicating the novelty of the current work compared to the already carried works.
2. In Section '2.3. Battery fabrication' the authors claim that ‘the addition of PMGC into the anode electrode did not yield favorable result’. A more detailed discussion is necessary to explain why.
3. The sale bar for the SEM images comprised in Figure 2 is not visible.
4. A quantitative discussion for the TGA results should be included.
5. XPS measurements are necessary for a complete structural characterization of PMGC composite.
6. What happens with the internal resistance of G9 battery in Figure 6?
7. There are few grammatical and typographical errors. Please check the manuscript and refine carefully.
For the reasons mentioned above, I support the publication of this manuscript after making the corresponding clarifications and modifications.
Sincerely,
The Reviewer
Reviewer 2 Report
Comments and Suggestions for Authors
Reviewer comments
This manuscript describes a strategy for using porous conductive polyaniline-modified graphene composites to enhance lithium battery performance. Though the chemistries used in this work are not novel and the strategy was reported a long time ago, the comprehensive investigation of this material in pouch cells is interesting. Therefore I recommend major revisions before the publication in Nanomaterials.
(1) Regarding the introduction section, the authors claimed that this strategy was rarely applied in lithium-based batteries. This is not correct. Please refer to the report (Journal of Power Sources, 2017, 340, 160; Materials Today Chemistry, 2020, 16, 100249), the strategy was reported 4-7 years ago but in different battery systems. The authors are requested to rewrite this part and cite these articles. Evaluating this strategy in pouch cells is quite new in this field.
(2) In Line 140 Page 4, the ratios selected for PMGC in cathodes and anodes are very low and in the narrow range. The authors need to explain what will happen once the ratio is over 1.0 wt%, and these elaborations can be added to the manuscript. Please make sure they are based on the weight or volume.
(3) In Line 157 Page 5, the reviewer was confused by the density change. It is not understandable that only adding a small amount of PMGC can result in a significant difference in densities. Error bars could be added to these values to contribute to the explanation.
(4) Based on Fig 2c, the formation of 3D structures on graphene sheets is quite interesting. Authors could elaborate more on the mechanism of 3D structure formation or use some nice diagrams.
(5) Authors tried to use PMGC as conductive additives to lower the internal resistance of cells and benefit cycling performance. However, the intrinsic electrical conductivity of PMGC is missing in this manuscript. This data is strongly required by the reviewer.
(6) In Fig. 6 and Fig. 7, it seems that cells G7 have the worst performance. Accordingly, the reviewer hypothesizes that this composite material is not stable once used in the anodes. Please double-check the electrochemical window of PMGC, which will decide what kind of battery systems are suitable and matchable.
(7) The cycling performance is quite promising. PMGC did help with the durability of Lithium-ion pouch cells. Authors need to provide more information on why PMGC can help with cyclability.
Comments on the Quality of English LanguageEnglish needs to be improved. There are several grammatical errors.
Reviewer 3 Report
Comments and Suggestions for Authors
The authors report on the enhancement of lithium-ion battery performance using a graphene composite additive modified with porous conductive polyaniline. Overall, this work has made some progress, but the current results cannot fully support the authors' conclusions due to the lack of data from control groups. It is recommended that the following major revisions be made before publication in Nanomaterials:
1)The authors conducted experiments to study the effects of different amounts of additive added to the cathode and anode on battery performance, thereby determining the optimal amount of additive. However, experiments with control groups, such as the use of pure graphene and pure PANi additives, were not conducted to effectively validate the advantages of the composite material. These experiments need to be supplemented.
2)The authors compared the effects of different amounts and forms of additives on battery internal resistance and electrochemical processes, as shown in Figures 6 and 11. It can be seen that the overall internal resistance and interfacial impedance of the battery are actually very small, even less than 0.1Ω. This value is so small compared to a large amount of reported lithium battery data that it can be considered negligible. Therefore, it raises the question of whether it is meaningful to optimize and improve performance if the battery's internal resistance is already so low.
3)In the experimental method section 2.1, the authors mentioned de-doping the composite material by introducing ammonia. It is well known that the high conductivity of polyaniline conductive polymers comes from its proton doping effect. Why did the authors choose to de-dope?
4)In the structural characterization section, the authors need to supplement the characterization results of pure PANI for comparison, such as FTIR and Raman spectroscopy. This is to determine whether the characteristic peaks appearing in the composite material belong to polyaniline and to confirm their interaction, thus determining whether a composite material has been formed rather than a simple mixture of the two.
5)The cycling performance has been improved, but what are the deeper underlying reasons behind this enhancement?
6)The authors also need to compare the progress made with existing results to confirm the effectiveness of the composite additive in enhancing performance. For example, the following related works should be referenced: “Improving Li reversibility in Li metal batteries through uniform dispersion of Ag nanoparticles on graphene”; “Multi-layered fluorinated graphene cathode materials for lithium and sodium primary batteries”; “One-step growth of the interconnected carbon nanotubes/graphene hybrids from cuttlebone-derived bi-functional catalyst for lithium-ion batteries”.
7)Although the authors provided an overview of the synthesis of PANI and graphene composite materials in the introduction, this overview deviates from the core of this work, namely "the effectiveness of their composite in enhancing lithium batteries." In fact, the authors should review and discuss the application of graphene, PANI, and their existing composites in lithium-ion batteries. Some of the latest relevant strategies and results should be cited: “Monodispersed SWNTs assembled coating layer as an alternative to graphene with enhanced alkali-ion storage performance.” “Solid-state mechanochemistry advancing two dimensional materials for lithium-ion storage applications: A mini review”.
8)Some figure captions are too simplistic and need to be more detailed, such as the caption for Figure 6.
Comments on the Quality of English Languagerelatively good
Round 2
Reviewer 2 Report
Comments and Suggestions for Authors
Concerns have been addressed properly.
Reviewer 3 Report
Comments and Suggestions for Authors
After the revisions, the author has addressed all the issues I raised, effectively enhancing the quality of the paper. Therefore, I recommend its publication in Nanomaterials.